# Association of Non-Alcoholic Fatty Liver Disease and Hepatic Fibrosis with Epicardial Adipose Tissue Volume and Atrial Deformation Mechanics in a Large Asian Population Free from Clinical Heart Failure

**DOI:** 10.3390/diagnostics12040916

**Published:** 2022-04-06

**Authors:** Yau-Huei Lai, Cheng-Huang Su, Ta-Chuan Hung, Chun-Ho Yun, Cheng-Ting Tsai, Hung-I Yeh, Chung-Lieh Hung

**Affiliations:** 1Department of Medicine, Mackay Medical College, New Taipei City 25245, Taiwan; garak1109@mmh.org.tw (Y.-H.L.); chsu007@gmail.com (C.-H.S.); hung0787@ms67.hinet.net (T.-C.H.); hiyeh@mmh.org.tw (H.-I.Y.); jotaro3791@gmail.com (C.-L.H.); 2Division of Cardiology, Department of Internal Medicine, Hsinchu MacKay Memorial Hospital, Hsinchu City 30071, Taiwan; 3MacKay Junior College of Medicine, Nursing, and Management, Taipei City 11260, Taiwan; 4Division of Cardiology, Department of Internal Medicine, MacKay Memorial Hospital, Zhongshan North Road, Taipei City 10449, Taiwan; 5Department of Radiology, MacKay Memorial Hospital, Zhongshan North Road, Taipei City 10449, Taiwan

**Keywords:** fatty liver, cardiovascular disease, fibrosis, epicardial fat, left atrial strain

## Abstract

Non-alcoholic fatty liver disease (NAFLD) and cardiovascular disease share several cardiometabolic risk factors. Excessive visceral fat can manifest as ectopic fat depots over vital organs, such as the heart and liver. This study assessed the associations of NAFLD and liver fibrosis with cardiac structural and functional disturbances. We assessed 2161 participants using ultrasound, and categorized them as per the NAFLD Fibrosis Score into three groups: (1) non-fatty liver; (2) fatty liver with low fibrosis score; and (3) fatty liver with high fibrosis score. Epicardial fat volume (EFV) was measured through multidetector computed tomography. All participants underwent echocardiographic study, including tissue Doppler-based E/e’ ratio and speckle tracking-based left ventricular global longitudinal strain, peak atrial longitudinal strain (PALS), and atrial longitudinal strain rates during systolic, early and late-diastolic phases (ALSR_syst_, ALSR_early_. ALSR_l__ate_). Larger EFV, decreased e’ velocity, PALS, ALSR_syst_, and ALSR_early_, along with elevated E/e’ ratio, were seen in all groups, especially in those with high fibrosis scores. After multivariate adjustment for traditional risk factors and EFV, fibrosis scores remained significantly associated with elevated E/e’ ratio, LA stiffness, and decreased PALS (β: 0.06, 1.4, −0.01, all *p* < 0.05). Thus, NAFLD is associated with LV diastolic dysfunction and subclinical changes in LA contractile mechanics.

## 1. Introduction

Obesity and metabolic disorders have long been recognized as global health issues. Non-alcoholic fatty liver disease (NAFLD), also known as metabolic-associated fatty liver disease (MAFLD) [1,2], is one of the most widespread forms of chronic liver disease. It represents a spectrum of conditions ranging from simple steatosis to non-alcoholic steatohepatitis, which can progress to various grades of fibrosis, cirrhosis, and hepatocellular carcinoma [3,4]. NAFLD has also been identified as a precipitating factor for early subclinical cardiac disorders since it shares several cardiometabolic risk factors with cardiovascular disease (CVD) [5]. Numerous studies have evaluated the pathological mechanisms connecting these two entities [6,7]. For example, NAFLD has been shown to induce hepatic insulin resistance and atherogenic dyslipidemia [8], which makes the affected individuals susceptible to premature atherosclerosis.

Excessive visceral fat can accumulate at various body sites besides the liver, most notably presenting as epicardial fat. As a biologically active source of pro-atherogenic cytokines that mediate systemic vascular inflammation and metabolic derangements [9], epicardial fat has been shown to influence left ventricular (LV) structure and function through mechanical or paracrine effects [10]. The term “cardiac steatosis” has been used to describe the exaggerated lipid deposition and elevated oxidative stress that up-regulates myocardial fibrosis and cellular apoptosis, leading to cardiomyopathy [11,12]. The release of a variety of pro-inflammatory and pro-fibrogenic mediators in this condition may play important roles in the pathophysiology of cardiac and arrhythmic complications. Furthermore, the surrounding ectopic fat depots may mechanistically impede diastolic filling due to the physical constraints on the epicardium [13].

Previous studies utilizing strain imaging have shown both cross-sectional and longitudinal association of NAFLD with subclinical myocardial remodeling and diastolic dysfunction [14,15]. It has been hypothesized that these changes may be mediated by the interaction between epicardial fat and cardiac structures. A few small histological studies have demonstrated a graded relationship between LV systolic dysfunction, epicardial fat thickness, and liver fibrosis severity in NAFLD, suggesting that systemic inflammation may also contribute to the development and progression of myocardial dysfunction [16,17]. Due to the cost of biopsy and procedural risks, alternative noninvasive tools have been developed to predict the fibrosis stage, including the NAFLD Fibrosis Score [18], which we employed in this study.

Whether or not NAFLD is an independent risk factor for cardiovascular mortality and other cardiovascular events has been studied, but remains controversial [19,20]. Relatively little is known about the underlying pathophysiological mechanisms behind the cardiac remodeling process in Asians with NAFLD. Therefore, we aim to investigate the potential impact of fatty liver and epicardial fat on various aspects of diastolic function and myocardial deformation mechanics in a large Asian population with normal LV ejection fraction, and who free from clinical heart failure (HF).

## 2. Materials and Methods

### 2.1. Study Population

We retrospectively examined individuals who participated in an ongoing cardiovascular health screening program between June 2009 and January 2013 at a tertiary medical center in Taipei, Taiwan. The original study setting and design have been published previously [21]. The baseline clinical information, medical history, symptoms/signs, and lifestyle patterns were obtained. Informed consent was waived for each participant owing to the retrospective study design. This study conformed to the principles outlined in the Declaration of Helsinki, and was approved by the local ethical board committee (18MMHIS180e, 8 January 2019). Subjects with a history of chronic viral hepatitis, liver cirrhosis, heavy alcohol consumption, any prevalent clinical HF, CVD (defined as a history of previous myocardial infarction, symptom-driven angioplasty, peripheral arterial disease, or cerebrovascular disease), and significant valvular diseases or cardiac arrhythmias were excluded from the analysis. Those with a high fibrosis risk score but diagnosed as non-fatty liver by abdominal ultrasound were also excluded, since it is unlikely for them to have liver fibrosis. (Figure 1).

### 2.2. Laboratory Data and Body Fat Assessment

All biochemical and laboratory parameters, including homeostatic model assessment for insulin resistance (HOMA-IR) and high-sensitivity C-reactive protein (Hs-CRP), were measured at a standardized central laboratory using a Hitachi 7170 Automatic Analyzer (Hitachi Corp., Hitachinaka, Ibaraki, Japan). Body fat composition was calculated by foot-to-foot bioelectrical impedance-based analysis (Tanita-305 Body-Fat Analyzer; Tanita Corp, Tokyo, Japan), which estimated the total body fat percentage.

### 2.3. Assessment of Fatty Liver and Fibrosis Score

Abdominal sonography was performed with a Toshiba Nemio SSA-550A instrument (Toshiba, Tochigi-ken, Japan) by hepatology specialists who were completely blinded to other laboratory results. The degree of fatty liver was graded semi-quantitatively according to the level of echoes arising from the hepatic parenchyma [22]. Since interpretation of fatty liver by abdominal ultrasound can be subjective, we defined subjects with at least moderate-degree fatty liver disease as significant NAFLD. By applying the low cut-off value (−1.455) of the NAFLD Fibrosis Score [18] (calculated as [−1.675 + 0.037 × age (years) + 0.094 × BMI (kg/m^2^) + 1.13 × hyperglycemia/diabetes mellitus (yes = 1, no = 0) + 0.95 × AST (U/L) to ALT (U/L) ratio − 0.013 × platelet count (10^−9^/L) − 0.66 × albumin (g/dL)]), the study population was categorized into three groups: (1) non-fatty liver; (2) NAFLD with low fibrosis score; and (3) NAFLD with high fibrosis score. APRI and FIB-4 scores were also calculated for comparison [23].

### 2.4. Assessment of Epicardial Fat

Multidetector computed tomography study was performed using a 16-slice scanner (Sensation 16, Siemens Medical Solutions, Forchheim, Germany) with 16 mm × 0.75 mm collimation, rotation time of 420 ms, and tube voltage of 120 kV. In one breath-hold, images were acquired from a level above the tracheal bifurcation to below the base of the heart, using prospective ECG-triggering at 70% of the R-R interval. From the raw data, the images were reconstructed with a standard kernel in 3-mm thick axial, non-overlapping slices, and a 25-cm field of view. Epicardial fat volume (EFV) was measured offline on a single workstation (Aquarius iNtuition Cloud, TeraRecon, SanMateo, CA, USA), using methods validated in previous studies [24].

### 2.5. Conventional Echocardiography and Diastolic Function Indices

Each participant underwent an extensive two-dimensional (2D) and tissue Doppler echocardiography with strain analysis. All assessments were performed by a single experienced sonographer blinded to the participants’ clinical information, using a commercially available ultrasound system equipped with a 2–4 MHz multifrequency transducer (Vivid 7; GE Medical System, Vingmed, Norway), in adherence with the American Society of Echocardiography guidelines [25]. Using the modified biplane Simpson’s method, the maximum values of left atrial volume (LAV) were presented in this study. All measurements were the average value derived from three consecutive cardiac cycles. Diastolic functional indices were assessed using transmitral pulsed-wave Doppler and tissue Doppler-derived mitral annular velocities. Systolic and early diastolic velocities (LV s’ and LV e’) were averaged from the basal septal and lateral LV segments at the mitral annulus level.

### 2.6. Two-Dimensional Speckle-Tracking Analysis Protocol

Speckle-tracking analysis was performed offline using 2D cardiac performance software (EchoPAC version 10.8; GE Vingmed Ultrasound, Norway). Semi-automated tracing of endocardial borders was performed at the end-diastolic frame, with minor manual adjustments to ensure optimal delineation. The LV global longitudinal strain (GLS) was calculated as the average peak global values derived from three LV apical planes of the 4-chamber, the 2-chamber, and long-axis views, as described in our previous published work [25]. Peak atrial longitudinal strain (PALS) and triphasic LA strain rates [systolic, early, and late diastolic atrial longitudinal strain rate (ALSR_syst_, ALSR_early_, and ALSR_late_, respectively)] were determined as the average values obtained from both apical 2- and 4-chamber views. The endocardial border of the LA was traced manually so that the LA appendage and pulmonary veins were excluded. LA stiffness (LA_stiff_) was derived from dividing E/e’ by PALS. To avoid confusion regarding the directionality of strain changes, the absolute values of GLS, ALSR_early_, and ALSR_late_ were reported. The inter-and intra-observer analysis of the LA and LV strain/strain rate components in our lab was reported in our previous work [21].

### 2.7. Statistical Analysis

Data were presented as mean (standard deviation) for continuous variables, and as proportions for categorical variables. One-way ANOVA was used to assess differences of anthropometric, metabolic, and echocardiography parameters between groups with post hoc paired comparisons using the Bonferroni correction. Fisher’s exact test was used to test differences between categorical data. Multiple linear regression models were used to assess the independent association of EFV and NAFLD Fibrosis Score with diastolic function and deformation parameters. For EFV, clinical covariates (age, sex, BMI, blood pressure, and clinical risk factors) were sequentially entered into the models. Since age and BMI are components of the NAFLD Fibrosis Score, they were omitted from the multivariate models for its analysis, while ALT was added into them. Statistical analyses were performed using the STATA statistical software package (Version 14. Stata Corp. College Station, Texas). A two-sided *p*-value of less than 0.05 was considered statistically significant.

## 3. Results

### 3.1. Baseline Demographics, Adiposity Measures, and Metabolic Profiles

Among the 2161 eligible study participants (mean age: 48.3 ± 9.9 years, 36.5% female), progressive epicardial fat burden, central obesity, stepwise increases in HbA1c, HOMA-IR, GOT, and decreases in platelet count were observed across all three groups (all *p* < 0.05). Both fatty liver groups shared similarly unfavorable lipid profiles and elevated Hs-CRP compared to the normal group. Patients with high fibrosis scores were also older and more likely to have hypertension, diabetes, and hyperlipidemia (Table 1). After excluding the youngest quartile from the normal group, differences in adiposity measures and biomarkers remained mostly unchanged (Appendix A).

### 3.2. Cardiac Structures, Diastolic Function and Strain Indices

Increasing LV wall thickness, LA and LV volumes, and LV mass (with and without indexation) was observed across all three groups (all *p* < 0.05). Progressive worsening diastolic function with graded reductions in LV e’ and elevated E/e’ ratio was also present (all *p* < 0.05). Most notably, stepwise reductions in PALS, ALSR_syst_, and ALSR_early_ across all three groups were observed (all *p* < 0.05). Compared to the normal group, both fatty liver groups had similar decreases in LV GLS (Table 2). After excluding the youngest quartile from the normal group, differences in LV geometry, LV e’, PALS, ALSR_syst_, ALSR_early_, and LA stiffness remained significant (Appendix A).

Box plots of EFV, PALS and LV GLS are further illustrated in Figure 2 and Figure 3.

Following multivariate adjustment, EFV remained strongly associated with elevated E/e’ ratio, and LA_stiff_ (β: 4.06, 41,14, both *p* < 0.05) and decreased LV e’, PALS, ALSR_syst_, and ALSR_early_ (β: −7.62, −2.48, −42.14, −31.94, all *p* < 0.05) (Table 3).

After similar adjustments, NAFLD Fibrosis Scores were weakly associated with EFV, yet still statistically significant (β: 0.01, *p* < 0.001). After further correction for EFV, the fibrosis score remained significantly associated with elevated E/e’ ratio, LA_stiff_ (β: 0.06, 1.4, both *p* < 0.001) and decreased LV e’, PALS, ALSR_syst_, and ALSR_early_ (β: −0.1, −0.01, −0.26, −0.4, all *p* < 0.05) (Table 4).

## 4. Discussion

In a large Asian population free from clinical HF and CVD, we demonstrated that NAFLD was associated with epicardial fat burden, systemic inflammation, insulin resistance, subclinical cardiac remodeling, diastolic dysfunction, and attenuated myocardial deformation. Moreover, diastolic function and most LA strain indices remained inversely correlated with NAFLD Fibrosis Score after correction for metabolic confounders and EFV.

Recently, the large population-based prospective CARDIA Study showed that NAFLD is longitudinally associated with subclinical LV remodeling, abnormal geometry, and impaired LV longitudinal strain after a five-year follow-up [15]. Several small studies have shown that NAFLD is associated with decreased LA strain indices [26,27]. There is a plethora of evidence that supports the adverse effects of NAFLD on diastolic function [28,29,30]. Although the pathogenesis of cardiac dysfunction in NAFLD is still unclear, insulin resistance, abnormal lipid metabolism, and systemic inflammation have been contributing factors. The cellular influx of free fatty acids may lead to myocardial lipid deposition, with consequent alterations in LV performance. Furthermore, hepatic steatosis is associated with hepatic insulin resistance, causing hyperglycemia and compensatory hyperinsulinemia, which may worsen both systemic and cardiac insulin resistance and subsequent myocardial dysfunction [31,32].

Several studies have demonstrated that HF with preserved ejection fraction (HFpEF) is highly prevalent in patients with underlying NAFLD [33,34]. The NAFLD Fibrosis Score is also associated with worse clinical outcomes among patients with HfpEF [35]. Thus, identifying predisposing factors for diastolic dysfunction is a pivotal first step toward implementing effective prevention strategies and treatment for HfpEF.

Consistent with our previous work based on epicardial fat thickness [36], one major finding of this study is that EFV was associated with LV diastolic dysfunction and LA contractile dysfunction independent of traditional risk factors and BMI. Previous epidemiological and clinical studies have consistently demonstrated that epicardial fat is related to the presence, severity, and recurrence of atrial fibrillation (AF) across various phenotypes [37]. Possible mechanisms include myocardial extracellular matrix turnover and fibrotic replacement, resulting in arrhythmogenic substrate formation [12]. Since LA strain is an established predictor of AF occurrence and recurrence [38], our findings may further support the usage of epicardial fat in AF risk evaluation. Interestingly, the association between EFV and LAV was attenuated after multivariate adjustment in our study, yet it did not appear to be masked by the effects of global obesity in terms of BMI.

Similarly, our study also showed that hepatic stiffness (predicted by NAFLD Fibrosis Score) was independently associated with LA contractile dysfunction, LA stiffness, and LV diastolic dysfunction, even after correction by EFV. This suggests a direct link between NAFLD severity and impaired LA/LV compliance, implying that hepatic fibrosis may be an additional risk factor in the development and progression of HfpEF and AF. Although ALSR_late_ is higher in the low fibrosis group compared to normal group, this is expected, since LA booster pump function could be augmented as a compensatory mechanism for decreased early filling [39]. Additional prospective studies are needed to further assess the putative mechanisms between hepatic histology and myocardial mechanics.

There are several limitations to our study. First, our study had a male sex predominance, which may be somewhat biased. Secondly, this survey is retrospective and cross-sectional, without a longitudinal follow-up or validation with clinical outcomes. Thirdly, our diagnosis of fatty liver was not based on liver biopsy, so it may not be accurate. Lastly, the study population comprises asymptomatic participants who underwent a primary cardiovascular health survey, and may not be fully representative of the broader general population in daily outpatient clinics.

## 5. Conclusions

NAFLD may play a significant role in developing HFpEF and AF, and this pathway may be mediated by epicardial fat accumulation. In a large Asian community, we demonstrated that hepatic fibrosis in NAFLD is independently associated with LV diastolic dysfunction, impaired LA deformation, and LA stiffness. More studies are required to determine the exact mechanisms between fatty liver and clinical HF.

## Figures and Tables

**Figure 1 diagnostics-12-00916-f001:**
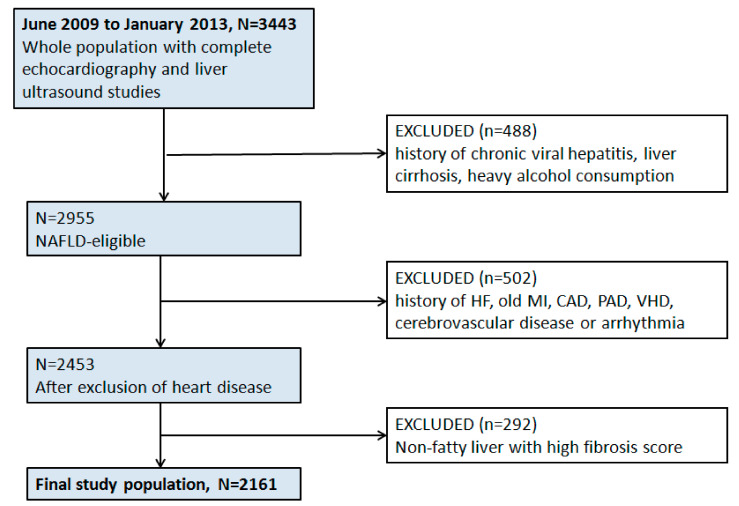
Study design and exclusion flowchart. Abbreviations: HF = heart failure, MI = myocardial infarction, CAD = coronary artery disease, PAD = peripheral artery disease, VHD = valvular heart disease.

**Figure 2 diagnostics-12-00916-f002:**
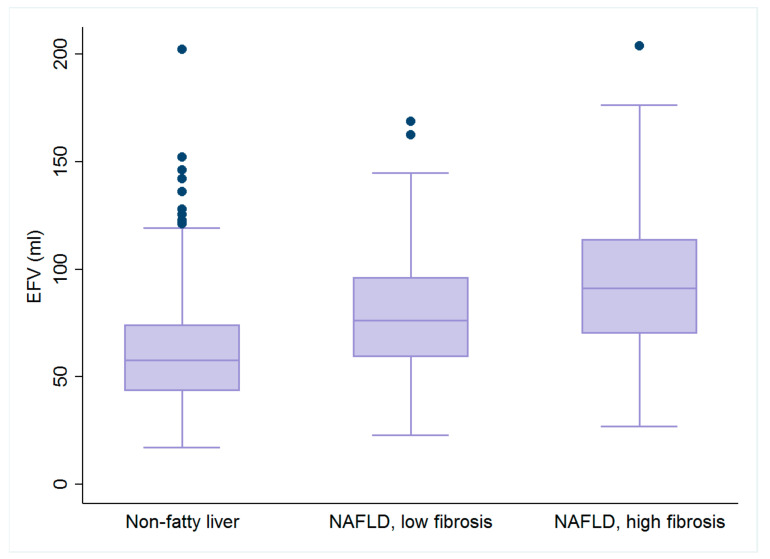
Box plots of epicardial fat volume (EFV). Graded increases in EFV were observed across all three groups (all *p* < 0.05).

**Figure 3 diagnostics-12-00916-f003:**
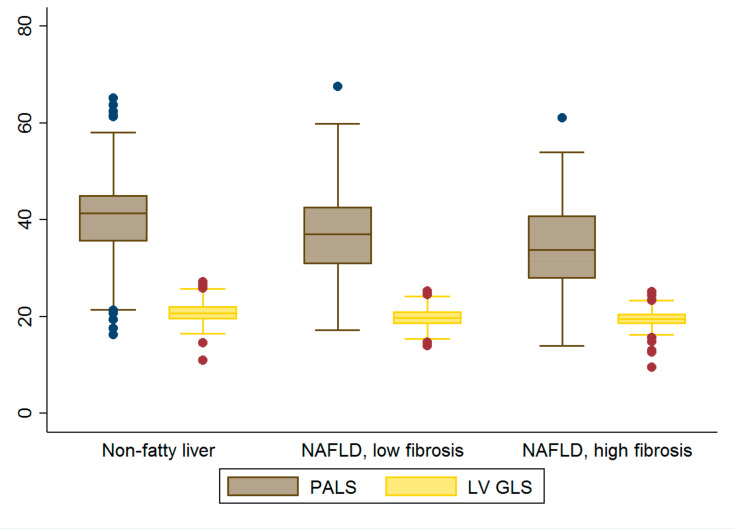
Box plots of LA and LV strain. Graded decreases in LA strain were observed across all three groups (all *p* < 0.05). Abbreviations: GLS= global longitudinal strain, PALS = peak atrial longitudinal strain.

**Table 1 diagnostics-12-00916-t001:** Baseline demographics, adiposity measures and biomarkers.

	Non-Fatty Liver	NAFLD, Low Fibrosis Score (<−1.455)	NAFLD, High Fibrosis Score (≥−1.455)	P_trend_
	N = 1019	N = 840	N = 302	
Age, years	46.32(9.96)	47.69(9.1) *	56.37(8.34) *†	<0.001
Female sex, %	502 (49.3%)	211 (25.1%) *	76 (25.2%) *	<0.001
NAFLD Fibrosis score	−2.76(0.79)	−2.67(0.79) *	−0.82(0.54) *†	<0.001
FIB-4 score	0.86(0.34)	0.93(0.29) *	1.35(0.48) *†	<0.001
APRI score	0.23(0.15)	0.37(0.26) *	0.42(0.28) *†	<0.001
SBP, mmHg	116.93(15.14)	123.86(16.42) *	129.21(16.03) *†	<0.001
DBP, mmHg	72.76(10.22)	78.04(10.21) *	80.45(10.06) *†	<0.001
**Adiposity measures**				
EFV, ml	65.02(25.5)	79.73(26.25) *	95.13(31.67) *†	<0.001
BMI, kg/m^2^	22.26(2.58)	25.69(3.17) *	27.21(3.53) *†	<0.001
WC, cm	77.86(7.9)	87.29(8.24) *	91.06(9.46) *†	<0.001
Body fat, %	23.84(6.26)	27.42(7.38) *	28.9(7.61) *†	<0.001
**Biomarkers**				
Fasting glucose, mg/dl	93.53(10.59)	100.15(18.17) *	118.17(32.1) *†	<0.001
HbA1c, %	5.54(0.44)	5.74(0.66) *	6.32(1.18) *†	<0.001
Fasting insulin, U/L	6.62(3.61)	10.01(6.16) *	10.86(5.81) *	<0.001
HOMA-IR	1.53(0.94)	2.5(1.78) *	3.23(2.05) *†	<0.001
Hs-CRP, mg/L	1.61(4.36)	2.42(4.16) *	2.67(3.73) *	0.001
Platelet, 10^9^/L	257.15(46.67)	243.96(48.03) *	207.87(32.99) *†	<0.001
PT-INR	1.04(0.04)	1.03(0.05) *	1.04(0.05)	0.03
GOT, IU/L	21.21(7.46)	26.11(11.3) *	28.1(14.57) *†	<0.001
GPT, IU/L	21.41(12.29)	36.42(24.24) *	34.44(21.94) *	<0.001
GGT, IU/L	20.55(18.49)	34.28(40.76) *	35.79(48.43) *	<0.001
Bil(d), mg/dL	0.21(0.07)	0.2(0.07)	0.22(0.08)	0.06
Bil(t), mg/dL	0.78(0.34)	0.81(0.36)	0.83(0.37)	0.06
Albumin, g/dL	4.52(0.25)	4.59(0.24) *	4.46(0.24) *†	<0.001
TC, mg/dL	200.91(36.12)	211.29(36.5) *	207.02(35.7) *	<0.001
TG, mg/dL	106.33(78.63)	167.19(95.16) *	170.79(111.15) *	<0.001
LDL-C, mg/dL	127.15(33.83)	140.58(33.38) *	136.89(31.84) *	<0.001
HDL-C, mg/dL	60.64(15.49)	48.77(12.45) *	47.96(11.41) *	<0.001
eGFR, mL/min/m^2^	91.78(16.6)	89.3(15.2) *	85.39(16.84) *†	0.001
**Comorbidities**				
Hypertension, %	82 (8%)	149 (17.7%) *	93 (30.8%) *†	<0.001
Diabetes, %	17 (1.7%)	65 (7.7%)	77 (25.5) *†	<0.001
Hyperlipidemia, %	43 (4.2%)	64 (7.6%)	36 (11.9%) *	<0.001

Data presented as mean (SD). *p*-value < 0.05 for comparisons against * Non-fatty liver, † Fatty liver with low fibrosis score, Abbreviations: SBP = systolic blood pressure, DBP = diastolic blood pressure, EFV = epicardial fat volume, BMI = body mass index, WC = waist circumference, HOMA-IR = homeostasis model assessment-insulin resistance, Hs-CRP = high-sensitivity C-reactive protein, PT-INR = prothrombin time-international normalized ratio, GOT = glutamic oxaloacetic transaminase, GPT = glutamate pyruvate transaminase, GGT = gamma-glutamyl transferase, Bil (d) = direct bilirubin, Bil (t) = total bilirubin, TC = total cholesterol, TG = triglyceride, LDL-C = low-density lipoprotein cholesterol, HDL-C = high-density lipoprotein cholesterol, eGFR = estimated glomerular filtration rate.

**Table 2 diagnostics-12-00916-t002:** Echocardiographic parameters.

	Non-Fatty Liver	NAFLD, Low Fibrosis Score (<−1.455)	NAFLD, High Fibrosis Score (≥−1.455)	P_trend_
	N = 1019	N = 840	N = 302	
LVST, mm	8.6(1.03)	9.14(0.96) *	9.54(1.02) *†	<0.001
LVPT, mm	8.6(0.94)	9.13(0.88) *	9.48(0.95) *†	<0.001
RWT	0.38(0.04)	0.39(0.04) *	0.4(0.04) *†	<0.001
LVEDV, mL	72.44(13.36)	77.26(12.48) *	80.75(11.04) *†	<0.001
LVEF, %	62.79(5.05)	62.16(5.2) *	62.34(5.08)	0.03
LVM, gm	129.66(29.83)	146.4(27.82) *	159.65(30.33) *†	<0.001
LVMi(BSA), gm/m^2^	72.32(13.73)	74.89(12.56) *	80.42(13.79) *†	<0.001
LVMi, gm/m^2.7^	33.66(7.13)	36.86(7.11) *	41.08(7.94) *†	<0.001
LAV, mL	26.69(8.92)	31.44(11.22) *	35.63(12.48) *†	<0.001
LAEF, %	58.9(10.59)	57.72(10.73)	56.94(10.48) *	0.006
**Diastolic function**				
DT, ms	196.21(37.2)	201.19(35.32) *	214.5(40.65) *†	0.001
IVRT, ms	87.76(13.49)	89.87(13.64) *	94.03(18.93) *†	<0.001
E/A	1.4(0.49)	1.19(0.37) *	1.04(0.36) *†	<0.001
LV e’, cm/sec	10.27(2.39)	9.11(2.04) *	7.85(1.88) *†	<0.001
LV s’, cm/sec	8.49(1.52)	8.32(2.07)	7.98(1.45) *†	<0.001
E/e’	6.71(2.21)	7.11(2.43) *	8.08(3.05) *†	<0.001
**Strain indices**				
LV GLS, %	20.85(1.91)	19.78(1.6) *	19.53(1.71) *	<0.001
PALS, %	40.23(7.39)	36.96(7.77) *	34.05(8.01) *†	<0.001
ALSR_syst_	1.78(0.38)	1.66(0.36) *	1.52(0.35) *†	<0.001
ALSR_early_	2(0.54)	1.66(0.49) *	1.35(0.45) *†	<0.001
ALSR_late_	2(0.49)	2.07(0.5) *	2.05(0.49)	0.006
LA_stiff_	0.17(0.08)	0.2(0.09) *	0.26(0.15) *†	<0.001

Data presented as mean (SD). *p*-value < 0.05 for comparisons against * Non-fatty liver, † Fatty liver with low fibrosis score. Abbreviations: LVST = left ventricular septal wall thickness, LVPT = left ventricular posterior wall thickness, RWT = relative wall thickness, LVEDV = left ventricular end-diastolic volume, LVM = left ventricular mass, LVMi = left ventricular mass index, LAV = left atrial volume, LAEF = left atrial emptying fraction, DT = deceleration time, IVRT = isovolumetric relaxation time, E/A = early-to-late diastolic mitral inflow velocity ratio, e’ = early-diastolic tissue Doppler velocity, s’ = systolic tissue Doppler velocity, GLS= global longitudinal strain, PALS = peak atrial longitudinal strain, ALSRsyst = atrial longitudinal strain rate-systolic phase, ALSRearly = atrial longitudinal strain rate-early diastolic phase, ALSRlate = atrial longitudinal strain rate-late diastolic phase, LAstiff = LA stiffness.

**Table 3 diagnostics-12-00916-t003:** Multiple linear regression analysis for association of EFV with diastolic function and deformation parameters.

	Pearson r	Univariate Model		Multivariate Model 1		Multivariate Model 2	
		β [95% CI]	*p*	β [95% CI]	*p*	β [95% CI]	*p*
LAV	0.22	0.76 [0.57, 0.95]	<0.001	0.18 [0.01, 0.36]	0.04	0.7 [−0.67, 2.07]	0.3
LVM	0.33	0.32 [0.25, 0.38]	<0.001	0.06 [−0.01, 0.14]	0.08	0.11 [−0.32, 0.55]	0.59
LV e’	−0.35	−4.75 [−5.66, −3.83]	<0.001	−1.33 [−2.36, −0.29]	0.01	−7.62 [−14.91, −0.32]	0.04
E/e’	0.19	2.23 [1.38, 3.07]	<0.001	0.71 [−0.07, 1.48]	0.08	4.06 [0.17, 7.95]	0.04
LV GLS	−0.27	−4.2 [−5.28, −3.12]	<0.001	−1.14 [−2.15, −0.13]	0.03	0.8 [−6.87, 8.48]	0.83
PALS	−0.34	−1.26 [−1.52, −1.01]	<0.001	−0.54 [−0.78, −0.3]	<0.001	−2.48 [−4.41, −0.55]	0.01
ALSR_syst_	−0.26	−19.51 [−24.8, −14.22]	<0.001	−8.57 [−13.31, −3.84]	<0.001	−42.14 [−68, −16.28]	0.002
ALSR_early_	−0.42	−23.37 [−26.97, −19.78]	<0.001	−8.05 [−12.38, −3.72]	<0.001	−31.94 [−58.13, −5.75]	0.02
LA_stiff_	0.33	102.47 [79.11, 125.83]	<0.001	41.51 [18.99, 64.04]	<0.001	41.14 [18.24, 64.05]	<0.001

Multivariate model 1: adjusted by age, sex, BMI; Multivariate Model 2: Model 1 + systolic blood pressure, total cholesterol, high-density lipoprotein, glomerular filtration rate, history of hypertension, diabetes, hyperlipidemia, and smoking.

**Table 4 diagnostics-12-00916-t004:** Multiple linear regression analysis for association of NAFLD Fibrosis Score with EFV, diastolic function, and deformation parameters *.

	Pearson r	Univariate Model		Multivariate Model 1		Multivariate Model 2	
		β [95% CI]	*p*	β [95% CI]	*p*	β [95% CI]	*p*
EFV	0.31	0.012 [0.01, 0.014]	<0.001	0.01 [0.007, 0.011]	<0.001	-	-
LAV	0.22	0.02 [0.017, 0.025]	<0.001	0.013 [0.01, 0.017]	<0.001	0.002 [−0.005, 0.01]	0.57
LVM	0.26	0.009 [0.007, 0.01]	<0.001	0.006 [0.004, 0.01]	<0.001	0.004 [0.001, 0.006]	0.002
LV e’	−0.42	−0.18 [−0.19, −0.16]	<0.001	−0.14 [−0.15, −0.12]	<0.001	−0.1 [−0.13, −0.06]	<0.001
E/e’	0.24	0.11 [0.1, 0.13]	<0.001	0.08 [0.06, 0.1]	<0.001	0.06 [0.02, 0.09]	<0.001
LV GLS	−0.14	−0.08 [−0.1, −0.05]	<0.001	−0.02 [−0.04, 0.003]	0.095	−0.01 [−0.05, 0.03]	0.64
PALS	−0.29	−0.036 [−0.03, −0.04]	<0.001	−0.025 [−0.03, −0.02]	<0.001	−0.01 [−0.02, −0.001]	0.02
ALSR_syst_	−0.25	−0.65 [−0.76, −0.55]	<0.001	−0.5 [−0.6, −0.4]	<0.001	−0.26 [−0.44, −0.07]	0.006
ALSR_early_	−0.44	−0.79 [−0.86, −0.72]	<0.001	−0.65 [−0.72, −0.55]	<0.001	−0.4 [−0.55, −0.25]	<0.001
LA_stiff_	0.33	3.34 [2.92, 3.76]	<0.001	2.39 [1.97, 2.81]	<0.001	1.4 [0.63, 2.17]	<0.001

Multivariate model 1: adjusted by sex, systolic blood pressure, ALT, total cholesterol, high-density lipoprotein, glomerular filtration rate, history of hypertension, diabetes, hyperlipidemia, and smoking; Multivariate Model 2: Model 1 + EFV; * Since age and BMI are components of the NAFLD Fibrosis Score, they are not added to the models.

## Data Availability

Owing to the local institutional regulation (which, in this study, was approved several years ago, and at that stage the authors did not apply for data spread or distribution out of the institution), together with the newly applied “Personal Information Protection Act” in Taiwan, the data will not be appropriate to be released in public place. The spread and data release will cause some concern from local ethical committees based on current institution regulations. Data are available from the “MacKay Memorial Hospital” Institutional Data Access/Ethics Committee for researchers who meet the criteria for access to confidential data. The contact information as follows: Mackay Memorial Hospital, Address: No. 92, Sec. 2, Zhongshan N. Rd., Taipei City 10449, Taiwan, Tel: 02-25433535#3486~3488, Email: mmhirb82@gmail.com (Institutional Review Board).

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
