# Peer review of "Association of Non-Alcoholic Fatty Liver Disease and Hepatic Fibrosis with Epicardial Adipose Tissue Volume and Atrial Deformation Mechanics in a Large Asian Population Free from Clinical Heart Failure"

_diagnostics, 2022, doi:10.3390/diagnostics12040916_

Round 1

Reviewer 1 Report

The manuscript (diagnotstics-1655405) entitled "Association of non-alcoholic fatty liver disease and hepatic fibrosis with epicardial adipose tissue volume and atrial deformation mechanics in a large Asian population free from clinical heart failure” provides reasonable and interesting results of association NAFLD and epicardial adipose tissues. Although this relation has been suggested well in several studies, it is still not possible to apply actively it to the treatment and management of patients based on an integrated understanding in the clinical setting. Thus, this paper is worthy that authors demonstrated the association between NAFLD fibrosis and cardiac functions, such as LA contractile dysfunction, LA stiffness, and LV diastolic dysfunction, even after correction by EFV. This study suggested the need for more in-depth research in this field. This paper is well-organized and written in an easy –to-understand manner for the reader. However, if the authors can provide answers some comments and questions and revise the paper, I recommend it is worth to publish in Diagnostics.

Major

  1. In addition to liver stiffness score by US you provided, please present biomarkers such as FIB-4 index, or APRI score. And, it is important to demonstrate prothrombin time (INR or %), and important values of liver function, and platelet count, which can be used as an indicator of portal hypertension.

  1. Other data showed that almost all of them had bad values in the high fibrosis group, but TC and LDL-C were higher in the low fibrosis group. How can you explain this?

  1. In addition, this part is similar to strain indices, and the index is significantly higher in the low fibrosis group. Please explain how to understand this in the discussion section.

Minor

Page 1 line 2: Coinedknown

Please check if the word is correct

Author Response

Thank you very much for your comments, which are very instructive, and very helpful to this manuscript and our future research. The responses to your comments are dictated below and all changes in manuscript were highlighted in yellow color.

  1. Regarding the comment “In addition to liver stiffness score by US you provided, please present biomarkers such as FIB-4 index, or APRI score. And, it is important to demonstrate prothrombin time (INR or %), and important values of liver function, and platelet count, which can be used as an indicator of portal hypertension.”

    Thank you very much for your comments and we’ve added FIB-4 and APRI scores, platelet count and INR in Table 1. The findings were summarized in Section 3.1.

  1. Regarding the comment “Other data showed that almost all of them had bad values in the high fibrosis group, but TC and LDL-C were higher in the low fibrosis group. How can you explain this?

    Thank you very much for your comments. Since cholesterol and LDL are primarily synthesized in the liver, people with relatively better liver reserve (low fibrosis group) might have higher TC and LDL-C levels. Moreover, the differences between these two groups were not statistically significant.

  1. Regarding the comment “In addition, this part is similar to strain indices, and the index is significantly higher in the low fibrosis group.”

     Thank you very much for your comments. As we showed in Table 2, only late-diastolic LA strain rate (ALSRlate) is higher in the low fibrosis group compared to normal group. Late-diastolic strain rate reflects LA booster pump function, which could be augmented as a compensatory mechanism for decreased early filling. We’ve inserted this explanation in the discussion section.

We’ve also corrected the wording error in page 1, line 2. Thanks for pointing this out.

Reviewer 2 Report

The authors report association of NAFLD and hepatic fibrosis with epicardial adipose tissue volume and atrial deformation mechanics in Asian population. There were many previous published results on the same sample and authors need to better explain the need for the  new one. Such limitations that NAFLD wasn’t a primary aim of the study and that the age difference between the groups should be mentioned and explained. I appreciate that multivariate analysis was conducted, but in view of above mentioned limitations there is still a concern on age effect. Would the conclusion remain the same if one quarter of the youngest non-NAFLD patients is excluded from analysis? This additional analysis could strengthen the validity of the study and I think that it should be added at least as supplementary information.

NAFLD score criteria for low and high score are not give, This should be also added to Table 1. Additionally, the reference 21 „The original study setting and design have been published previously [21]“ leads to article where again another reference is given to find a study design. Replace this reference with the reference that leads to article where the whole description can be find.

Author Response

  1. Regarding the comment “Would the conclusion remain the same if one quarter of the youngest non-NAFLD patients is excluded from analysis? This additional analysis could strengthen the validity of the study and I think that it should be added at least as supplementary information.”

     Thank you very much for your comments. We’ve added this analysis in Supplementary Table 1 and 2. Differences in adiposity measures and biomarkers remained mostly unchanged. Most notably, differences in LV e’, PALS, ALSRsyst, ALSRearly and LA stiffness remained significant. These findings were summarized in Section 3.1 and 3.2.

  1. Regarding the comment “NAFLD score criteria for low and high score should be added to Table 1. Additionally, the reference 21 leads to article where again another reference is given to find a study design.

    Thank you very much for your comments and we’ve corrected these accordingly.

The above descriptions are the responses to your comments and suggestions.